# Nationwide increases in anti-SARS-CoV-2 IgG antibodies between October 2020 and March 2021 in the unvaccinated Czech population

Pavel Piler [1,8], Vojtěch Thon [1,8 ✉], Lenka Andrýsková[1], Kamil Doležel[2], David Kostka[3], Tomáš Pavlík [4,5], Ladislav Dušek[4,5], Hynek Pikhart [1,6], Martin Bobák[1,6], Srdan Matic [7] & Jana Klánová[1]

## Abstract

**Background** The aim of the nationwide prospective seroconversion (PROSECO) study was to investigate the dynamics of anti-SARS-CoV-2 IgG antibodies in the Czech population. Here we report on baseline prevalence from that study.

**Methods** The study included the first 30,054 persons who provided a blood sample between October 2020 and March 2021. Seroprevalence was compared between calendar periods, previous RT-PCR results and other factors.

**Results** The data show a large increase in seropositivity over time, from 28% in October/November 2020 to 43% in December 2020/January 2021 to 51% in February/March 2021. These trends were consistent with government data on cumulative viral antigenic prevalence in the population captured by PCR testing – although the seroprevalence rates established in this study were considerably higher. There were only minor differences in seropositivity between sexes, age groups and BMI categories, and results were similar between test providing laboratories. Seropositivity was substantially higher among persons with history of symptoms (76% vs. 34%). At least one third of all seropositive participants had no history of symptoms, and 28% of participants with antibodies against SARS-CoV-2 never underwent PCR testing.

**Conclusions** Our data confirm the rapidly increasing prevalence in the Czech population during the rising pandemic wave prior to the beginning of vaccination. The difference between our results on seroprevalence and PCR testing suggests that antibody response provides a better marker of past infection than the routine testing program.

**Plain Language Summary**

The nationwide prospective seroconversion (PROSECO) study is monitoring the presence of anti-SARS-CoV-2 IgG antibodies in the Czech population. The presence of these antibodies is indicative of a previous infection with SARS-CoV-2. Here, we report initial results from 30,054 participants in this study that were recruited between October 2020 and March 2021, during the second wave of the COVID pandemic. The data indicates that the percentage of the population infected with SARS-CoV-2 is higher than estimates based on official data on cumulative PCR testing for presence of the virus. At least one third of participants with antibodies did not have symptoms of COVID and 28% had not undergone PCR testing. This study demonstrates the importance of monitoring the presence of anti-SARS-CoV-2 IgG antibodies to accurately assess the proportion of the population that has been infected over time.

[1] RECETOX, Faculty of Science, Masaryk University, Kotlarska 2, 611 37 Brno, Czech Republic. [2] QualityLab Association, Evropská 846/176a, Prague, Czech Republic. [3] Health Insurance Company of the Ministry of the Interior of the Czech Republic, Vinohradská 2577/178, 130 00 Prague, Czech Republic. [4] Institute of Biostatistics and Analyses, Faculty of Medicine, Masaryk University, Kamenice 3, 625 00 Brno, Czech Republic. [5] Institute of Health Information and Statistics of the Czech Republic, Palackého náměstí 4, 128 01 Prague, Czech Republic. [6] Department of Epidemiology & Public Health, University College London, 1 – 19 Torrington Place, London WC1E 6BT, UK. [7] World Health Organization (WHO), Country Office in the Czech Republic, Rytířská 31, 110 00 Prague, Czech Republic. [8] These authors contributed equally: Pavel Piler, Vojtěch Thon ✉email: vojtech.thon@recetox.muni.cz

The COVID-19 pandemic is caused by severe acute respiratory syndrome coronavirus 2 (SARS-CoV-2)[1]. This virus stimulates a rapid seroconversion to IgG antibodies in symptomatic[2,3] as well as asymptomatic subjects[4,5]. Therefore, the memory IgG antibodies against SARS-CoV-2 can serve as a specific long-term biomarker of previous SARS-CoV-2 infection[6] as well as an appropriate marker for monitoring the persistence of antibodies against SARS-CoV-2 after vaccination[7].

The SeroTracker dashboard, which systematically monitors and synthesises findings from hundreds of global SARS-CoV-2 serological studies[8], shows a lack of nationwide population-based seroprevalence studies in Central and Eastern Europe, compared to several seroprevalence studies in other parts of Europe[9–11]. While some existing studies have examined general population samples (ranging between 959[12] and 105,651[13] participants), most have been restricted to hotspots[14–17]. These studies adopted different sampling and analytical strategies: some have used household and community sampling[9,17] while others focused on biosamples from blood donors[18] or residual sera in laboratories[10,12]. While the majority of these studies employed serological tests, point-of-care antibody assays with lower sensitivity and specificity were also used[13,19]. A review by Rostami et al. reported seroprevalence for Europe between 0.66% and 5.27% until August 2020[20]. Thus far, only a small number of nationwide population-based studies have been published covering the period between November 2020 and March 2021, i.e. the second wave of the epidemic in Europe[21–23].

In the Czech Republic, the first COVID-19 cases were confirmed on 1 March 2020. The first epidemic wave in spring 2020 was relatively modest, with only 9301 cases recorded by the end of May 2020 (86.9 confirmed cases per 100,000 persons over three months). The governmental restrictions were dismantled in the summer; this resulted in a worsening of the epidemic situation in autumn 2020. At the peak of the second wave, over 10,000 new cases were diagnosed every day (over 100 confirmed cases per 100,000 persons daily) and the Czech Republic ranked among the countries with the greatest burden of COVID-19 in Europe and in the world. As of 31 May 2021, the Czech Republic had a cumulative total of 1,661,787 cases confirmed (15,528 per 100,000 persons) and the highest cumulative incidence of COVID-19 among countries with ≥1,000,000 inhabitants[24].

To address the lack of regional data and to investigate the dynamics of seroprevalence of anti-SARS-CoV-2 IgG antibodies in the region, a nationwide prospective seroconversion study (PROSECO) was initiated in the Czech Republic. The aim was to study the dynamics of seroconversion in three six-month-long periods: (1) the second epidemic wave (unvaccinated population); (2) the mass vaccination period; and (3) the post-vaccination period. Here, we describe the study design and the results of the first phase of the study (October 2020 to March 2021). Our findings show a rapid increase in seroprevalence in the Czech population during the rising pandemic wave prior to the beginning of vaccination and suggest that antibody response provides a better marker of past infection than the routine testing program.

## Methods

**Study design and study population.** The Czech PROSECO study investigates seroconversion after SARS-CoV-2 infection (or vaccination), as well as the decline of memory IgG antibodies against SARS-CoV-2 over time. The study has been planned as a series of three consecutive phases, each lasting six months, with participant enrolment taking place in the first phase. The first phase of the study, described here, was completed just before the launch of a massive nationwide vaccination programme.

All clients of the second-largest health insurance company in the Czech Republic were sent a written invitation to participate in the study. On September 20 the first batch of invitation letters was sent to all clients (>18 years old) and the first 30,054 persons who provided blood samples between 1 October 2020 and 31 March 2021 were included in the study population. The presence of COVID-19 symptoms on the day of study enrolment was used as the sole exclusion criterion. The participants paid <20% (i.e. a negligible sum) of an antibody test cost.

The participants were invited to visit their local blood collection centres managed by one of the five different chains of clinical laboratories incorporated to QualityLab association covering the entire area of the Czech Republic. A 3 ml blood sample was collected in a separating gel vacuum tube. Serum samples were distributed at +4 °C and subsequently analysed in central laboratories within 12 h. Leftover sera were stored in a biobank for future use.

Participants completed a questionnaire that included personal information (such as residence address, date of birth, sex), anthropometric data, self-reported results of previous RT-PCR tests (if performed), history of symptoms compatible with COVID-19, and records of COVID vaccination.

Informed consent forms were obtained from all study participants during each wave of the data collection. An ethics committee approval of all aspects of data collection, as well as of the secondary data analysis, was obtained from the ELSPAC ethics committee under reference number (C)ELSPAC/EK/5/2021.

**Detection of IgG antibodies against SARS-CoV-2.** CE-marked serological tests were performed in accredited clinical laboratories. Antigen-specific humoral immune response was analyzed by detection of IgG antibodies against the spike protein using commercial immunoassays LIAISON SARS-CoV-2 S1/S2 IgG (DiaSorin, Saluggia, Italy) and SARS-CoV-2 IgG II Quant (Abbott, Sligo, Ireland). Testing was conducted on the LIAISON XL (DiaSorin, Saluggia, Italy) and on the Alinity (Abbott, Lake Forest, IL, USA) respectively. Samples were tested individually and reported according to the manufactures´ criteria.

**Statistics and reproducibility.** The primary aim of this study was to determine the seroprevalence of the adult Czech population. We estimated seroprevalence rates and 95% confidence intervals. We also standardized the seroprevalence rates to age and sex, using the Czech population as a standard[25]. We used a multivariate Poisson regression model with a robust error variance to evaluate the differences in the seroprevalence rate between study periods, participant characteristics and testing providers and to compare SARS-CoV-2 antibody reactive individuals to non-reactive individuals. Differences in prevalence were expressed as prevalence rate ratios (PRRs) between each category and reference group. We used standard descriptive statistics to characterize the study data set.

Population data on COVID-19 were obtained from the Czech Central Information System of Infectious Diseases (ISID), which includes records of all consecutive patients with COVID-19 in the Czech Republic identified and confirmed by laboratory testing. ISID data are routinely collected in compliance with Act No. 258/2000 Coll. On the Protection of Public Health and are publicly available in aggregated and anonymized form of open or authenticated data sets. All analyses were conducted using Stata version 15.1 (StataCorp, College Station, Texas 77845 USA).

## Results

Between October 2020 and March 2021, 30,054 individuals consented to participate in the study and their results were

**Table 1 Overview of seroprevalence estimates between October 2020 and March 2021**

| Period | SARS-CoV-2 serology test result | | Seroprevalence (95% CI) standandardized by age and sex for national population | p value |
|---|---|---|---|---|
| | Total n | Positive n (%) | | |
| October 2020– November 2020 | 3626 | 1025 (28.3%) | 27.9% (26.1–29.7) | p < 0.001 |
| December 2020– January 2021 | 6880 | 2984 (43.4%) | 42.2% (40.8–43.5) | |
| February 2021– March 2021 | 19,548 | 10,052 (51.4%) | 51.0% (50.3–51.8) | |
| Total | 30,054 | 14,061 (46.8%) | 46.4% (45.8–47.0) | |

CI confidence interval.

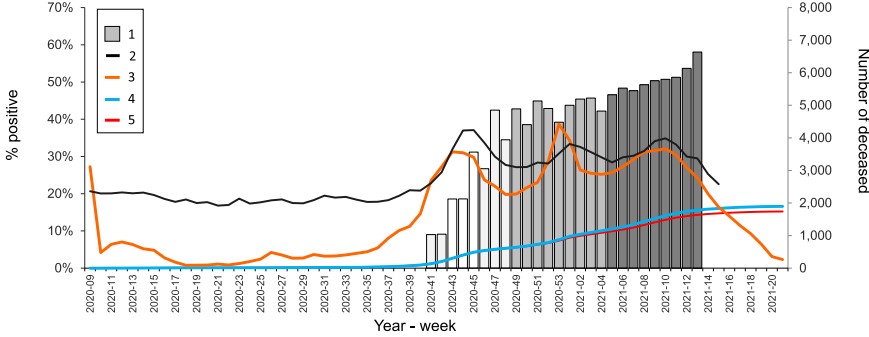

**Fig. 1 Weekly seroprevalence of SARS-CoV-2 antibodies in pandemic wave before vaccination in the Czech Republic.** Dynamics of the COVID-19 pandemic in the Czech Republic and seroprevalence in the first phase of the PROSECO study between October 2020 and March 2021. (1) % of persons with SARS-CoV-2 antibodies in the PROSECO study; (2) Number of deaths from all causes according to the Czech Statistical Office (all ages); (3) % of positive PCR tests among all PCR tests provided (all ages); (4) Cumulative % of persons positively tested for SARS-CoV-2 (PCR or antigen) among persons aged 18+; (5) Cumulative % of persons positively tested for SARS-CoV-2 (PCR only) among persons aged 18+.

included in the final analyses. The seroprevalence overview is shown in Table 1. During the entire study period, 14,061 (46.8%) of the 30,054 individuals were tested positive for anti-SARS-CoV-2 IgG antibodies. After adjusting for national population distribution with regards to sex and age, this corresponds to 46.4% (95% CI: 45.8–47.0). The seroprevalence increased over time, along with the rise of the epidemic curve during the study period (Fig. 1). In the first two months (October–November 2020), we estimated an overall seroprevalence of 27.9% (95% CI: 26.1–29.7). This estimate increased to 42.2% (95% CI: 40.8–43.5) in next two months (December 2020 to January 2021), and to 51.0% (95% CI: 50.3–51.8) in February and March 2021.

Table 2 shows differences in seroprevalence rates between various population groups. Seropositivity was slightly higher in the 18–29 and 50–59 age groups than among those aged 30–39, 40 –49 and 60+. Rates were similar in men and women. Individuals who reported having previously one or more COVID-19 symptoms had higher seroprevalence when compared to asymptomatic participants (76.9% vs. 33.7%); the history of COVID-19 symptoms was associated with higher seropositivity in all three two-month periods (all p-values p < 0.001). A higher seropositivity prevalence was also observed in persons with BMI > 30 kg/m² (Supplementary Data 1). No statistically significant differences were observed in the pattern of seropositivity levels by sex (p = 0.086) and participating laboratories (range of p values is between 0.111 and 0.765).

Table 3 shows that RT-PCR tests were previously performed on 15,799 individuals (52.6% of the study sample) and of these, 62.0% were found SARS-CoV-2 seropositive. Seropositivity was also detected in 28.4% of individuals who were never tested using RT-PCR and in 29.3% of those who reported negative RT-PCR test results. On the other hand, 15.8% of the individuals reporting positive RT-PCR test results were found seronegative.

Figure 1 shows the weekly seroprevalence in this study along with national cumulative data on SARS-CoV-2 positivity (blue line). While the absolute levels are very different, the temporal trends established using the two data sources are virtually identical, with a very rapid increase (over fivefold) between October 2020 and March 2021. Both positive PCR tests (orange line) as well as cumulative PCR values (red/blue line) demonstrate the rapid spread of the virus in the Czech population, which resulted in a dramatic increase in all-cause mortality throughout this period (Supplementary Data 2).

## Discussion

This study demonstrated a rapid increase in anti-SARS-CoV-2 IgG antibodies in the unvaccinated Czech adult population during the second COVID-19 pandemic wave between October 2020 and March 2021. We found a massive increase in seroprevalence of IgG antibodies against SARS-CoV-2 from 9 to 51.0% over the course of six months (Fig. 1). This reflects high viral load in the population with minimal anti-epidemic measures in the Czech Republic in autumn 2020.

Seroprevalence was low at the beginning of the study (October 2020), which is in agreement with similar studies performed in other countries until September 2020. Data available for a similar period from the Netherlands, France, Spain, UK, and Italy, for instance, suggested seroprevalence between 3.4 and 11.6%[9,10,12,13,18]. By the end of March 2021, the seropositivity rate reached 50% in our study subjects. The dynamics of seroprevalence in our study corresponded with nationwide Czech government data.

Only small differences in the seroprevalence of anti-SARS-CoV-2 IgG antibodies were observed among various population groups. Similar to the previous studies[9,14,26], no difference was found between men and women (Table 2). In agreement with

**Table 2 Prevalence rate ratios (PRRs) and 95 % confidence intervals for seroprevalence of IgG antibodies to SARS-CoV-2 by study periods, sex, age groups and COVID symptoms in PROSECO study participants estimated by multivariate Poisson regression**

| | October 2020 - November 2020 n=3626 | | | | December 2020-January 2021 n=6880 | | | | February 2021-March 2021 n=19,548 | | | | All study periods n=30,054 | | | |
|---|---|---|---|---|---|---|---|---|---|---|---|---|---|---|---|---|
| | Particip.[a] n (%) | Seropositive[b] (%) | PRR (95% CI) | p-value | Particip.[a] n (%) | Seropositive[b] (%) | PRR (95% CI) | p-value | Particip.[a] n (%) | Seropositive[b] (%) | PRR (95% CI) | p-value | Particip.[a] n (%) | Seropositive[b] (%) | PRR (95% CI) | p-value |
| **Study periods** | | | | | | | | | | | | | | | | |
| 10/2020-11/2020 | - | - | - | - | - | - | - | - | - | - | - | - | 3626 (12.1%) | 28.3% | 1 | - |
| 12/2020-01/2021 | - | - | - | - | - | - | - | - | - | - | - | - | 6880 (22.9%) | 43.4% | 1.41 (1.34–1.49) | **<0.001** |
| 02/2021-03/2021 | - | - | - | - | - | - | - | - | - | - | - | - | 19,548 (65.0%) | 51.4% | 1.67 (1.59–1.76) | **<0.001** |
| **Sex** | | | | | | | | | | | | | | | | |
| Male | 1424 (39.3%) | 27.7% | 1 | - | 2756 (40.1%) | 41.8% | 1 | - | 7612 (38.9%) | 51.1% | 1 | - | 11,792 (39.2%) | 46.1% | 1 | - |
| Female | 2202 (60.7%) | 28.7% | 1.00 (0.90–1.11) | 0.996 | 4124 (59.9%) | 44.4% | 1.04 (0.98–1.09) | 0.171 | 11,936 (61.1%) | 51.7% | 1.02 (0.99–1.04) | 0.223 | 18,262 (60.8%) | 47.2% | 1.02 (1.00–1.04) | 0.086 |
| **Age groups** | | | | | | | | | | | | | | | | |
| 18–29 | 318 (8.8%) | 31.1% | 1 | - | 582 (8.5%) | 43.1% | 1 | - | 1660 (8.5%) | 52.5% | 1 | - | 2560 (8.5%) | 47.7% | 1 | - |
| 30–39 | 662 (18.3%) | 24.8% | 0.80 (0.66–0.98) | **0.034** | 976 (14.2%) | 36.9% | 0.83 (0.74–0.94) | **0.002** | 2621 (13.4%) | 47.9% | 0.90 (0.85–0.95) | **<0.001** | 4259 (14.2%) | 41.8% | 0.88 (0.83–0.92) | **<0.001** |
| 40–49 | 1354 (37.3%) | 28.3% | 0.91 (0.76–1.09) | 0.299 | 2262 (32.9%) | 43.8% | 0.97 (0.88–1.07) | 0.548 | 5515 (28.2%) | 51.4% | 0.95 (0.90–1.00) | **0.032** | 9131 (30.4%) | 46.1% | 0.95 (0.91–0.99) | **0.020** |
| 50–59 | 842 (23.2%) | 31.0% | 0.98 (0.82–1.19) | 0.866 | 1845 (26.8%) | 47.5% | 1.06 (0.96–1.17) | 0.219 | 5020 (25.7%) | 53.7% | 0.99 (0.94–1.04) | 0.574 | 7707 (25.6%) | 49.8% | 1.00 (0.96–1.05) | 0.884 |
| 60+ | 450 (12.4%) | 26.2% | 0.87 (0.70–1.08) | 0.218 | 1215 (17.7%) | 41.6% | 0.97 (0.87–1.08) | 0.572 | 4732 (24.2%) | 50.6% | 0.94 (0.90–1.00) | **0.032** | 6397 (21.3%) | 47.2% | 0.95 (0.90–0.99) | **0.022** |
| **COVID symptoms** | | | | | | | | | | | | | | | | |
| Asymptomatic | 2266 (62.5%) | 18.1% | 1 | - | 3941 (57.3%) | 29.4% | 1 | - | 11,119 (56.9%) | 38.5% | 1 | - | 17,326 (57.7%) | 33.7% | 1 | - |
| Symptomatic | 767 (21.2%) | 62.3% | 3.47 (3.13–3.85) | **<0.001** | 2080 (30.2%) | 75.0% | 2.61 (2.47–2.76) | **<0.001** | 5873 (30.0%) | 79.5% | 2.09 (2.03–2.15) | **<0.001** | 8720 (29.0%) | 76.9% | 2.27 (2.22–2.33) | **<0.001** |
| Unknown | 593 (16.4%) | 23.1% | 1.21 (0.91–1.60) | 0.191 | 859 (12.5%) | 30.8% | 1.11 (0.97–1.28) | 0.139 | 2556 (13.1%) | 43.3% | 1.17 (1.10–1.25) | **<0.001** | 4008 (13.3%) | 37.6% | 1.17 (1.11–1.24) | **<0.001** |

PRR prevalence rate ratio, COVID symptoms symptoms compatible with COVID-19, CI confidence interval.
P < 0.05 was considered significant (in bold).
[a]Number of participants.
[b]Percentage of seropositive participants.

**Table 3 SARS-CoV-2 seroprevalence based on RT-PCR status**

| | Number of seropositive participants n (%) | Number of seronegative participants n (%) | All participants n (%) |
|---|---|---|---|
| RT-PCR status (n = 30,054) | | | |
| Never performed | 3554 (28.4%) | 8950 (71.6%) | 12,504 (100%) |
| Performed | 9799 (62.0%) | 6000 (38.0%) | 15,799 (100%) |
| Unknown | 708 (40.4%) | 1043 (59.6%) | 1751 (100%) |
| RT-PCR results (n = 15,799) | | | |
| Negative | 1736 (29.3%) | 4199 (70.7%) | 5935 (100%) |
| Positive | 7846 (84.2%) | 1475 (15.8%) | 9321 (100%) |
| Unknown | 217 (40.0%) | 326 (60.0%) | 543 (100%) |

other publications, our results indicated some differences in prevalence between age groups[26,27]. The lowest rate of seropositivity was observed in the 30–39 age group, followed by the 40–49 and 60+ age groups. The highest risk was estimated for the youngest age group 18–29 and for people in their fifties (50–59). Similar to previously published studies such as Ward et al. in the UK[13], we also established a higher risk for the obese subpopulation. These findings can be partially explained by sociological data from a national survey monitoring behaviour of various population groups and their adherence to the restrictions during the pandemic (zivotbehempandemie.cz).

Most seropositive participants reported history of more than one symptom related to the SARS-CoV-2 infection. On the other hand, around 30% of seropositive subjects were asymptomatic, which is consistent with other studies[9,13]. The ratio of asymptomatic and symptomatic participants differs among study phases: twice as many seropositive participants were asymptomatic in the third vs. in the first phase. This could be explained by exposure to low doses of the virus repeatedly over time, which could lead to seroconversion without the manifestation of symptoms (Table 2).

It should be also noted that 28% of participants who never underwent RT-PCR testing for SARS-CoV-2 were seropositive. Furthermore, 29% of seropositive participants who were tested by RT-PCR had previously had negative RT-PCR test results. Neither of these groups was visible in the official government statistics. On the other hand, 16% of seronegative participants previously reported a positive PCR test (Table 3). It may be expected that the mucosa-associated lymphoid tissue of these individuals dealt with the infection without triggering a systemic immune response[28].

To the best of our knowledge, the Czech PROSECO study is the largest study of the seroprevalence of anti-SARS-CoV-2 IgG antibodies in Central and Eastern Europe. According to the WHO seroepidemiological investigation protocol[29], the major strengths of our study are its size and coverage, its launch prior to the beginning of the vaccination period and the on-going longitudinal follow-up. In contrast to other larger studies, we assessed antibody levels in the harmonized QualityLab network of accredited clinical laboratories. As a result, we observed good agreement between seroprevalence results reported from five participating test providers. Some deviations observed in one lab in one sampling period were most probably random variation due to small number of analysed samples in that particular laboratory. Highly sensitive and specific chemiluminescence serological immunoassays were applied targeting the receptor-binding domain (RBD) of the spike protein of SARS-CoV-2 virus. Since previous studies have shown that the SARS-CoV-2 spike protein RBD is the target of neutralizing antibodies, this molecule has become a particularly interesting target for serological testing and vaccine development[30–32]. Our study also has the advantage of including data from serological testing as well as self-reported RT-PCR test results. This allows us to explore the spread of infection more precisely in the nationwide study population.

The study also has important limitations. The main weakness is the self-selection of study participants. Potential participants were contacted through their health insurance company (newsletter, letter) and using online advertising. Similar to other national seroprevalence studies, children were excluded for obvious reasons (informed consent, biological tissue requirement). The first 30,054 volunteers reporting to the QualityLab network were then enroled in the study, after which the recruitment procedure was discontinued. As a result, the true response rate cannot be determined and, despite its large size, this study sample may not reliably represent the entire population, since the participants were self-selected volunteers rather than a random sample selected from a specific sampling frame. It is likely that persons at higher risk of past or current infection are overrepresented in this study, which would lead to an overestimation of the absolute seropositivity prevalence rate. As we currently do not have information about socio-economic status and geographical location of study participants, we were unable to further assess the potential selection bias and to evaluate geographical differences in seropositivity. Crucially, however, the temporal trend in seropositivity in this study sample closely reflects the cumulative PCR positivity trend in national data, thus suggesting the good internal validity of our results.

Compared to the national statistics, female and middle-aged participants (40–59 years) were over-represented in our study at the expense of males, respondents from 18 to 39 years, and those over 60 years of age. The discrepancy in age profile can be partially attributed to the fact that the study participants were clients of one particular health insurance company, which has a larger representation of middle-aged clients compared to the general population. We also excluded institutionalised persons, which resulted in an underestimation of the oldest age category living in retirement/nursing homes, and individuals who were in prisons or monasteries at the time of the study. Availability of socio-economic and geographical data in future phases of the study will allow us to better interpret the findings and assess regional trends.

In March 2021, the national vaccination program started, aiming at vaccinating the entire population against SARS-CoV-2 by the end of September 2021. The second phase of the PROSECO study was therefore launched to follow-up on the study participants throughout the most intense vaccination period (i.e. from April to September 2021). The third phase, planned from October 2021 to March 2022, will investigate the dynamics and persistence of IgG SARS-CoV-2 antibodies. This longitudinal follow-up, which thus includes pre-vaccination, vaccination and postvaccination periods, together with the potential linkage of resulting data to morbidity and mortality data from the national registers will further increase the importance and impact of this surveillance[33].

In conclusion, the results of the first phase of the PROSECO study indicate that a significant part of the Czech adult population is no longer immunologically naïve to the SARS-CoV-2 infection. The massive increase of seroprevalence demonstrates the extensive exposure of the Czech population to the SARS-

CoV-2 virus between October 2020 and March 2021. The number of seropositive participants who never underwent RT-PCR testing demonstrates the importance of serological population-based studies describing the spread and exposure to the virus in the population over time. Seroconversion and positivity in specific antibody testing is a powerful additional diagnostic surrogate marker of SARS-CoV-2 antigenic challenge.

## Data availability

All data generated during the first phase of the PROSECO study is presented in this article. Anonymised data can be made available from the corresponding author upon request once all study phases are completed and data validated. Release of data is a subject of approval of the Ethical and Scientific boards of the PROSECO study.

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

## Acknowledgements

The PROSECO study was sponsored by the Prevention Programme of the Health Insurance Company of the Ministry of the Interior of the Czech Republic. The RECETOX Research Infrastructure was supported by the Ministry of Education, Youth and Sports of the Czech Republic (LM2018121), and V.T. and P.P. from the CETOCOEN PLUS project of ESIF (CZ.02.1.01/0.0/0.0/15_003/0000469). The authors acknowledge funding from the Ministry of Education, Youth and Sport of the Czech Republic/ESIF (CZ.02.1.01/0.0/0.0/17_043/0009632). This work was supported by the European Union's Horizon 2020 research and innovation programme under grant agreement nos. 857560 and 857487. This publication reflects only the authors' view, and the European Commission is not responsible for any use that may be made of the information it contains. We thank all collaborating nurses, laboratories from the QualityLab association, and administrative personnel and especially the 30,054 participants who invested their time and provided samples and information for this study.

## Author contributions

V.T., P.P., L.A. and J.K. were responsible for the design of the study, L.A. for ethics, and D.K., L.A. and V.T. for communication to potential participants. K.D., P.P. and L.A. were responsible for the study operation, coordination of data acquisition and logistics, and K.D. for coordination and quality management of participating laboratories. T.P., P.P. and L.D. were in charge of statistical analyses and table and figure design. The first draft was written by P.P. and V.T., M.B., H.P. and S.M. provided expertise in epidemiology. All authors contributed to data interpretation, critically reviewed the first draft, approved the final version and agreed to be accountable for the work.

## Competing interests

The authors declare no competing interests.
