## [Peer Review File · Communications Medicine]

Reviewers' comments:

Reviewer #1 (Remarks to the Author):

Thank you for inviting me to review this interesting and well written manuscript. The study reported is of baseline results from a large antibody prevalence study in adults in the Czech Republic. This is novel in that it includes a large nationwide population survey in a region with limited prevalence data.

The authors report a dramatic increase in seroprevalence between the earliest and latest recruited participants (Oct 2020 to March 2021), i.e., a change over time which was consistent with case-based surveillance which showed a wave of infection during that period. However, the scale of the increase in prevalence in the study (from 9% to 51%) is far higher than data from case-based surveillance would suggest. I think that there are some questions to be addressed about this.

- The overall design is a national, longitudinal study with repeated measures over time to directly measure seroconversion. However, the data reported here are only from the baseline, with no repeat measures. Therefore, I think it is a bit misleading to report findings as a seroconversion rate. The data are cross-sectional with recruitment taking place over 6 months and looking at time trends among the participants but not seroconversion. While this cross-sectional data may permit an estimate of seroconversion rate but is not a direct measure and several caveats need to be applied.
- For this to be a reasonable estimate of seroconversion in the population, the sample at each time point would need to be similar to avoid bias in relation to differential participation over time. There is insufficient information provided about recruitment to assess this. Table 2 reports characteristics for the three time periods and shows some differences, with a higher proportion of older people over time, a higher proportion reporting COVID symptoms, and a higher proportion with overweight or obesity. It would be interesting to know if there were differences by occupation, social class, household size and other known determinants of risk.
- Recruitment was through inviting everyone registered with a health insurance group to take part, and the first 30,000 to respond became the sample; participants had to pay towards the cost of the test and therefore presumably this was promoted as something that was valuable for people. It is quite possible that this offer (and willingness to pay) would be more attractive to people who think that they had had COVID-19, and therefore the estimates of prevalence would be inflated.

The authors reflect on these limitations in the discussion, but I think a more thorough assessment of the sample and reporting would help with interpretation.

Reviewer #2 (Remarks to the Author):

Overall, the authors present a coherent and concise study describing the seroconversion of volunteers in the population of Czech Republic, which is underreported in public records and provides an important data set for retrospective analysis of the COVID19 pandemic and warrants publication. The sample size of >30K is impressive and the statistical analyses are appropriate.

A few questions to address;

Line 104: The authors state that 'no major differences' were found. Can you include the statistical method and p value for that measurement in the text for ease?

Line 113: It is interesting that during Oct 2020 to March 2021, lockdowns were in full effect yet the country saw the highest seroconversion. Can the authors speculate on how this could have happened in the light of how public health policies may or may not have been effective?

Line 148: Do you have morbidity and mortality data on these volunteers? Did either of these factors change over time normalized to population?

Line 219: Do you know if seroconversion was detected with anti-S or anti-N or both? The former will detect vaccine + natural, the latter will detect only natural infection. This will be important for phase 3 study

Line 233: did you see any geographic trends in your data based on residence information from the questionnaire?

Reviewer #3 (Remarks to the Author):

Thank you for the opportunity to review your manuscript titled "Rapid dynamics of seroconversion of anti-SARS-CoV-2 IgG antibodies in the Czech unvaccinated population: nationwide prospective seroconversion (PROSECO) study"

The stated objective is to describe the dynamics of SARS-CoV-2 seroconversion on a population-based level.

The data presented is important to the COVID-19 field and to the public health community overall. The study design and data collection as well as the statistical analysis is sound and the conclusions are valid.

The manuscript is nicely written, concise and easy to understand. In my view, it should be published.

I have only minor comments and should be easy to address before publication.

- 1) please provide legend to your tables and charts
- 2) please clarify the p-value given in Table 1: which comparison is significant and which is not
- 3) Introduction, your statement starting line 72: mitigation measures obviously impact SARS-CoV-2 transmission trends. However, if you want to make such a strong statement you either need to give a reference or more background +/- data. Otherwise I suggest phrasing the paragraph a bit more carefully.
- 4) seroprevalence results from the 5 different test providers are usually pretty close with the exception of the December/January time period where provider 1 provides a much lower number compared to the other 4. This should be addressed/discussed.
- 5) Discussion of differences in seroprevalence in different age groups (Discussion starting line 130) I understand that these are hypothesis, however, they seem extremely random. 18-29 yo do not

respect rules and 50-59 yo did not use home office essentially. Here you should at least provide some reason why you hypothesize this. is there any data that adherence to control measures was lower in the younger age group? Is there any data that 50-59yo did not use home office? If not I would prefer if this paragraph is phrased more carefully.

COMMSMED-21-0440-T

Rapid dynamics of anti-SARS-CoV-2 IgG antibodies in the Czech unvaccinated population: nationwide prospective (PROSECO) study

Pavel Piler; Vojtěch Thon*; Lenka Andrášková; Kamil Doležel; David Kostka; Tomáš Pavlík ; Ladislav Dušek; Hynek Píkhart; Martin Bobák; Srdan Matic; Jana Klánová

COMMUNICATIONS MEDICINE

Reviewer #1 (Remarks to the Author):

Thank you for inviting me to review this interesting and well written manuscript. The study reported is of baseline results from a large antibody prevalence study in adults in the Czech Republic. This is novel in that it includes a large nationwide population survey in a region with limited prevalence data.

The authors report a dramatic increase in seroprevalence between the earliest and latest recruited participants (Oct 2020 to March 2021), i.e., a change over time which was consistent with case-based surveillance which showed a wave of infection during that period. However, the scale of the increase in prevalence in the study (from 9% to 51%) is far higher than data from case-based surveillance would suggest. I think that there are some questions to be addressed about this.

1. The overall design is a national, longitudinal study with repeated measures over time to directly measure seroconversion. However, the data reported here are only from the baseline, with no repeat measures. Therefore, I think it is a bit misleading to report findings as a seroconversion rate. The data are cross-sectional with recruitment taking place over 6 months and looking at time trends among the participants but not seroconversion. While this cross-sectional data may permit an estimate of seroconversion rate but is not a direct measure and several caveats need to be applied.

Response:

Thank you for this comment. We agree that the term seroconversion is potentially misleading since we did not have repeated data on initially seronegative subjects at the beginning of the study. We replaced the term “seroconversion” by “seroprevalence” throughout the manuscript where appropriate.

2. For this to be a reasonable estimate of seroconversion in the population, the sample at each time point would need to be similar to avoid bias in relation to differential participation over time. There is insufficient information provided about recruitment to assess this. Table 2 reports characteristics for the three time periods and shows some differences, with a higher proportion of older people over time, a higher proportion reporting COVID symptoms, and a higher proportion with overweight or obesity. It would be interesting to know if there were differences by occupation, social class, household size and other known determinants of risk.

Response:

We agree with this comment, and we attempted to address this important issue in the Discussion. Unfortunately, we do not have information on socioeconomic indicators from the baseline, although these data are currently being collected during the cohort follow up and will be added to the data analysis in the next phase of the PROSECO project. We expanded the discussion on this limitation. (Lines 193-196 and lines 206-207 in the document with the highlighted changes)

3. Recruitment was through inviting everyone registered with a health insurance group to take part, and the first 30,000 to respond became the sample; participants had to pay towards the cost of the test and therefore presumably this was promoted as something that was valuable for people. It is

quite possible that this offer (and willingness to pay) would be more attractive to people who think that they had had COVID-19, and therefore the estimates of prevalence would be inflated.

The authors reflect on these limitations in the discussion, but I think a more thorough assessment of the sample and reporting would help with interpretation.

Response:

Thank you and, again, we agree that such a selection bias is plausible. As with the point above, we discussed this potential bias in the Discussion in the original manuscript. As noted above, we do not currently have information of many individual characteristics, including socioeconomic status. Prompted by the reviewer, we expanded this discussion on the potential selection bias. (Lines 193-196 and lines 206-207 in the document with the highlighted changes).

Reviewer #2 (Remarks to the Author):

Overall, the authors present a coherent and concise study describing the seroconversion of volunteers in the population of Czech Republic, which is underreported in public records and provides an important data set for retrospective analysis of the COVID19 pandemic and warrants publication. The sample size of >30K is impressive and the statistical analyses are appropriate. A few questions to address.

1. Line 102: The authors state that 'no major differences' were found. Can you include the statistical method and p value for that measurement in the text for ease?

Response:

Thank you. We specified the statistical methods, and we added the p-value in the revised manuscript. (Lines 107-111 and 260-265)

2. Line 113: It is interesting that during Oct 2020 to March 2021, lockdowns were in full effect, yet the country saw the highest seroconversion. Can the authors speculate on how this could have happened in the light of how public health policies may or may not have been effective?

Response:

Thank you for the suggestion. We have discussed this issue briefly in the original submission, but we have expanded the discussion on this point. (Lines 128-129)

3. Line 148: Do you have morbidity and mortality data on these volunteers? Did either of these factors change over time normalized to population?

Response:

We agree that this would be a valuable information and it is feasible to get it in the future as the PROSECO participants agreed to the linkage of PROSECO data to the national registers in their informed contents. However, it is a lengthy procedure due to the ethical and data safety procedures and we were unable to access data on morbidity and mortality of the study participants so far. We are currently in discussion with the Institute of Health Information and Statistics about retrospective

linkage of the data. We expanded Discussion on the information about a lack of data in study phase 1 (Lines 213-214).

4. Line 237: Do you know if seroconversion was detected with anti-S or anti-N or both? The former will detect vaccine + natural, the latter will detect only natural infection. This will be important for phase 3 study.

Response:

Thank you for this important comment for the future study phases. We used CE-marked serological tests for analysing IgG antibodies against S1/S2 part of SARS-CoV-2. We added this information to the Methods section. (Line 252)

5. Line 230: did you see any geographic trends in your data based on residence information from the questionnaire?

Response:

Thank you for this suggestion. The data are currently not in the form that we could reliably classify participants in relation to their permanent residence. This information will be in the next phases and geographical trends will be an important focus of future analyses. (Lines 193-196 and lines 206-207)

Reviewer #3 (Remarks to the Author):

Thank you for the opportunity to review your manuscript titled "Rapid dynamics of seroconversion of anti-SARS-CoV-2 IgG antibodies in the Czech unvaccinated population: nationwide prospective seroconversion (PROSECO) study".

The stated objective is described the dynamics of SARS-CoV-2 seroconversion on a population-based level.

The data presented is important to the COVID19 field and to the public health community overall. The study design and data collection as well as the statistical analysis is sound, and the conclusions are valid.

The manuscript is nicely written, concise and easy to understand I in my view should be published.

I have only minor comments and should be easy to address before publication.

1) please provide legend to your tables and charts

Response:

This is now done in the revised manuscript.

2) please clarify the p-value given in Table 1: which comparison is significant, and which is not

Response:

Thank you for this comment. Table 1 shows overall p-value for heterogeneity in prevalence rates between the time periods, and statistical significance of differences between period 2 vs. period 1 and period 3 vs period 1 are shown in table 2.

3) Introduction, your statement starting line 70-72: mitigation measures obviously impact SARS-CoV-2 transmission trends. However, if you want to make such a strong statement you either need to give a reference or more background +/- data. Otherwise, I suggest phrasing the paragraph a bit more carefully.

Response:

Thank you for your suggestion, we have re-written this sentence in a more cautious way in the revised manuscript. (Lines 73-76)

4) Seroprevalence results from the 5 different test providers are usually pretty close with the exception of the December/January time period where provider 1 provides a much lower number compared to the other 4. This should be addressed/discussed.

Response:

Thank you for this comment. Provider 1 reported a relatively small number of analyses compared to the others. This disproportion can be a reason for observed differences between providers. As suggested above, other comparisons don't show such deviations. We added a comment in the Discussion (Lines 174-177).

5) Discussion of differences in seroprevalence in different age groups (Discussion starting line 130) I understand that these are hypothesis, however, they seem extremely random. 18-29 y do not respect rules and 50-59 yo did not use home office essentially. Here you should at least provide some reason why you hypothesize this. Is there any data that adherence to control measures was lower in the younger age group? Is there any data that 50-59yo did not use home office? If not I would prefer if this paragraph is phrased more carefully.

Response:

Thank you for these important questions. We rewrote this paragraph in the revised manuscript providing a reference to the available data from the national sociological survey monitoring various aspects of the pandemic in the Czech Republic (zivotbehempandemie.cz). (Lines 141-143)

Reviewers' comments:

Reviewer #1 (Remarks to the Author):

I think that the manuscript is improved with revision but I suggest further clarifications and editing. Some of the editing suggestions are to remove subjective superlatives which I think detract from the data which speak for themselves.

1. The title still includes the reference to seroconversion, and to rapid dynamics which I don't understand. I would suggest a more factual title, for example "Anti-SARS-CoV-2 IgG antibodies in the unvaccinated population in the Czech Republic: a nationwide study"
2. Line 32 also refers to the seroconversion study. I would suggest adding a new sentence. "Here we report on baseline prevalence from that study."
3. Line 38 I think "large" would be better than "dramatic".
4. Line 50: consider removing "dramatically" and "massively"
5. Line 75: consider removing "dramatic"
6. Lines 107 to 110: no need to include the p values which are in the table.
7. Line 132: "oscillating" is incorrect. I think you are referring to variation between studies.
7. The discussion remains long with some speculation, for example line 141 - 154 on variation by age - these theories are not based on the data or cited material.

Reviewer #3 (Remarks to the Author):

Thank you for the revision. My concerns have been addressed.

COMMSMED-21-0440-T

Anti-SARS-CoV-2 IgG antibodies in the unvaccinated population: Czech nationwide study

Pavel Piler; Vojtěch Thon*; Lenka Andrášková; Kamil Doležel; David Kostka; Tomáš Pavlík ; Ladislav Dušek; Hynek Pikhart; Martin Bobák; Srdan Matic; Jana Klánová

COMMUNICATIONS MEDICINE

Reviewer #1 (Remarks to the Author):

I think that the manuscript is improved with revision, but I suggest further clarifications and editing. Some of the editing suggestions are to remove subjective superlatives which I think detract from the data which speak for themselves.

The title still includes the reference to seroconversion, and to rapid dynamics which I don't understand. I would suggest a more factual title, for example "Anti-SARS-CoV-2 IgG antibodies in the unvaccinated population in the Czech Republic: a nationwide study".

Response:

We agree with this comment and, we revised the title to be more understandable.

Line 32 also refers to the seroconversion study. I would suggest adding a new sentence. "Here we report on baseline prevalence from that study."

Response:

Thank you for this comment. We added this sentence to the Abstract.

Line 38 I think "large" would be better than "dramatic".

Response:

Thank you for this comment. We replaced the word "dramatic" by the word "large".

Line 50: consider removing "dramatically" and "massively"

Response:

Thank you for this comment. We removed the words "dramatically" and "massive".

Line 75: consider removing "dramatic"

Response:

Thank you for this comment. We removed the word "dramatic" in this part of the text.

Lines 107 to 110: no need to include the p values which are in the table.

Response:

Thank you. The p-value was added to the revised manuscript because of the request and suggestion of the second reviewer.

Line 132: "oscillating" is incorrect. I think you are referring to variation between studies.

Response:

We agree with this comment and, we removed this word from the text.

The discussion remains long with some speculation, for example line 141 - 154 on variation by age - these theories are not based on the data or cited material.

Response:

Thank you for the suggestion. We agree with your comment, and we shortened this part in the revised manuscript.